



# Long-term Total Water Storage Change from a SAtellite Water Cycle (SAWC) reconstruction over large south Asian basins

Victor Pellet[1,2], Filipe Aires[1,2], Fabrice Papa[3], Simon Munier[4], and Bertrand Decharme[4]

[1]Laboratoire d'Études du Rayonnement et de la Matière en Astrophysique et Atmosphères, Observatoire de Paris, Paris, France.
[2]Estellus, Paris, France.
[3]Laboratoire d'Études en Géophysique et Océanographie Spatiales, IRD, Toulouse, France.
[4]CNRM-Université de Toulouse, Météo-France, CNRS, Toulouse, France

**Correspondence:** Victor Pellet(victor.pellet@obspm.fr)

**Abstract.** The Total Water Storage Change (TWSC) over land is a major component of the global water cycle, with a large influence on climate variability, sea level budget and water resources availability for human life. Its first estimates at large-scale were made available with GRACE observations for the 2002-2016 period, followed since 2018 by the launch of GRACE-FO mission. In this paper, using an approach based on the water mass conservation rule, we proposed to merge satellite-based observations of precipitation and evapotranspiration along with *in situ* river discharge measurements to estimate TWSC over longer time periods (typically from 1980 to 2016), compatible with climate studies. We performed this task over five major Asian basins, subject to both large climate variability and strong anthropogenic pressure for water resources, and for which long term record of *in situ* discharge measurements are available. Our SAtellite Water Cycle (SAWC) reconstruction provides TWSC estimates very coherent in terms of seasonal and interannual variations with independent sources of information such as (1) TWSC GRACE-derived observations (over the 2002-2015 period), (2) ISBA-CTRIP model simulations (1980-2015), and (3) multi-satellite inundation extent (1993-2007). This analysis shows the advantages of the use of multiple satellite-derived data sets along with *in situ* data to perform hydrologically coherent reconstruction of missing water component estimate. It provides a new critical source of information for long term monitoring of TWSC and to better understand their critical role in the global and terrestrial water cycle.

## 1 Introduction

Continental freshwater, excluding ice caps, represents only few percents of the total amount of water on Earth. Nevertheless they have a major impact on terrestrial environment and human life and activities and play a very important role in climate variability. Thus, understanding and predicting continental water storage variations is a topic of great importance for climate research, global water cycle studies (IPCC, 2014) and water resource management. In particular, the Total Water Storage Change (TWSC), comprising of all water mass variations from surface waters (wetland, floodplains, lakes, rivers and man-made reservoirs), soil moisture, snow pack, glaciers and groundwater, is of high interest because it represents a good indicator of potential long-term water cycle (WC) modifications related to natural or anthropogenic factors (Rodell et al., 2018).





Therefore, monitoring long-term spatio-temporal changes in continental freshwater storage has become fundamental. This question is particularly important for regions such as South Asia that experienced drastic changes over the last decades. The region includes some of the worlds largest rivers (Figure 1), originating in the Himalayas and crossing densely inhabited areas of the Indian subcontinent or South-East Asia, where changes in freshwater availability (Babel and Wahid, 2008) might threaten food and security for more than a billion people (Shamsudduha and Panda, 2019; Wijngaard et al., 2018).

Given the limited availability of *in situ* data in the region, satellite observations are unique to monitor the dynamic of terrestrial waters (Tiwari et al., 2009; Papa et al., 2015; Salameh et al., 2017) and analyzed their recent large-scale changes (Rodell et al., 2009; Asoka et al., 2017; Khaki et al., 2018). In particular, since 2002, the GRACE mission (Tapley et al., 2004) monitors the mass gravity field variation and provides an estimation of the TWSC at the monthly scale (for the period 2002-2016), followed since 2018 by the Gravity Recovery and Climate Experiment Follow-On (GRACE-FO). However, GRACE data time span is still too limited to study the long-term behavior of the WC related to climate changes or to human practices.

Recently, several studies have attempted to monitor the WC and provide independent estimates of TWSC using satellite observations (Lawford et al., 2007; Sheffield et al., 2009; Wood and Schaepman, 2009; Pan et al., 2012; Rodell et al., 2015; Munier and Aires, 2017). These analyses potentially allow new opportunities for the WC monitoring over long time-records in regions with limited access to *in situ* measurements. The use of satellite data to study the WC is however not straightforward. (1) Various datasets exist for the same geophysical variable and (2) they all have uncertainties (systematic bias and random errors), which lead to (3) the inconsistency between estimate of the same variable or among the variable estimates of the WC (Pellet et al., 2018a). No singular estimate can be considered as perfect and many authors preferred to combine various available products (Sheffield et al., 2009; Sahoo et al., 2011; Azarderakhsh et al., 2011; Lorenz et al., 2014). For that purpose, some have focused on the water conservation equation:

$$\Delta S \quad = \quad P - E - D \tag{1}$$

where $\Delta S$ is the TWSC, $P$ the Precipitation, $E$ the Evaporation, and $D$ the Discharge (expressed in mm/month, area-normalised). This closure of the WC budget allows to better constrain the integration of the datasets. For instance, Pan et al. (2012); Zhang et al. (2016) used an assimilation approach with Kalman filtering in a land surface model to derive a coherent analyze of the four terrestrial water variables ($P$, $E$, $D$ and $\Delta S$). Rodell et al. (2015) used variational 3D-VAR strategy to optimize the water cycle estimates at the global and annual scales.

Other approaches perform this integration independently from any model, which allows the integrated datasets being interesting for the calibration/validation of the model. Pan and Wood (2006); Aires (2014) have presented several methodologies to integrate coherently different hydrological datasets based on a budget closure. Munier et al. (2014) applied one of them (Aires, 2014) over the Mississippi basin using remote sensing observation for $P$, $E$ and $\Delta S$ and *in situ* measurment for $D$. The optimal integration is based on, first, a Simple Weighting (SW) average, then, a closure Post-Filtering (PF). The SW+PF method improved the WC components estimate compared to *in situ* observation. The uncertainties of integrated product are reduced compared to the original datasets, the coherency is improved, and the residuals of the WC budget closure are decreased. Furthermore, the authors have develloped a calibration approach based on the integrated product, able to correct each original



estimate in an independent way. This calibration led to a significative reduction of the budget residual (see also Pellet et al. (2018b)). It was shown in Munier et al. (2014) that when three out of four WC components in Eq. (1) are available, the reconstruction of the missing one can be attempted. This is possible if the signal-to-noise ratio is sufficient: discharge reconstruction

60    was not possible in Munier et al. (2014) but the TWSC could be obtained in a very simple way, with results quality comparable to a complex assimilation into a hydrologic model.

In this study, we propose to use this methodology to reconstruct the long-term evolution of the TWSC over large South Asian basins, based on satellite and *in situ* measurements and no hydrological model. We denote "SAWC" this SAtellite Water Cycle reconstruction. Section 2 introduces the tools used in this study, including a description of the region, the data sets used

and the methodology. Section 3 presents the results and evaluations while section 4 draws the conclusions and perspectives.

## 2    Materials and methods

### 2.1    Basins

Table 1 lists the basins considered in this study. They are also represented in Fig. 1. They were defined by first choosing river discharge (D) *in situ* measurement stations close to the sea, over the major Himalayan rivers, with a long-enough time record.

The HydroSHEDS topography (Lehner et al., 2006) was then used to determine the drainage area and basin delineation. The basins were selected based on: (1) their spatial domain needs to be large enough compared to the spatial resolution of the GRACE instrument, (2) the river discharge measurements need to cover the GRACE period (2002-2015).

Five basins were chosen:

– *Mekong*: The Mekong Delta is one of the largest deltas in the world. It is a vast plain (55000 km$^2$) mostly lower than 5 m

above sea level. Due to the seasonal variation in water level, the area presents extensive wetlands. The Mekong Delta region that represents only 12% of the total Vietnam area, allows  50% of the annual rice production (up to three harvests per year on some provinces),  50% of the fisheries, and  70% of the fruit production. Furthermore, questions related to oceanic water intrusions, change of agriculture practices (e.g. number of rice harvest in one year), dam construction, ground water pumping and resulting land depletion, all have an impact on the TWSC and would therefore benefit from

its monitoring.

– *Ganges and Brahmaputra*: The Ganges-Brahmaputra is the major river basin of the Indian Sub-Continent supplying more than 700 millions people. It covers an area of 1.7 million km$^2$, at the crossroad of Bangladesh, India, China, Bhutan and Nepal and is the third largest freshwater provider to the world's oceans (after the Amazon and the Congo rivers) with a high influence on the regional climate. The basin is seasonally subject to the monsoon and faces strong

climate variability between drought and floods periods. Furthermore, water management is an issue because of the increasing needs the population and the demands for the industry and agriculture sectors. Thus, the freshwater supply leads to an over-abstraction of groundwater stock during dry-season, and then to a rapid fall of groundwater tables.





- *Godavari*: The Godavari River is the second longest river of India after Ganges, covering a total drainage area of 312000 km$^2$ and accounting for nearly 9.5% of the total geographical area of the country. It flows for a length of about 1465
30   km, from its origin near the Arabian Sea before outfalling in to the Bay of Bengal, crossing several states of India. The basin receives its maximum rainfall during the southwest monsoon, from June to September. The major part of basin is covered with agricultural land accounting up to 60% of the total area, while 3.5% of the basin are covered by water bodies. Godavari basin faces several hydroclimatic problems with a large portion of the basin being prone to drought, while flooding problems are common in its lower reaches and its coastal areas are cyclone-prone.

- *Irrawaddy*: Running over a length of 2100km mainly within the boundaries of Myanmar, the Irrawaddy River is the most important river of the country. The basin takes up the northwestern part of the Indochina Peninsula, with its source on the south slopes of the Himalayas Mountains and emptying into the Andaman Sea of the Indian Ocean. The river basin area covers more than 400000 km$^2$ and collects 2/3 of the surface water volume of Myanmar. It is subject to a tropical and subequatorial monsoon climate and its hydrological regime, similarly to other large rivers of south Asia, is fed with
water on the slopes of the Himalayas Mountains, mainly from rains during the southwest monsoon period and melt water of glaciers. The river is vital for human activities, water supply, and irrigation and hosts a high biodiversity. It is prone to extreme events, such as floods from very heavy monsoon rains or extreme weather events like cyclones and severe droughts and under climate change impacts, the region is facing major challenges for water resources.

## 2.2   Datasets

### 2.2.1   Datasets used in the integration

The datasets presented in this section will be used in the integration process to obtain an optimised description of the WC over the Himalayan basins. Most of them are satellite products. Only global satellite products have been considered. In order to integrate them, the datasets have been projected onto a common 0.25° spatial resolution grid using a conservative interpolation (Jones, 1999), and re-sampled at the monthly scale.

*Precipitation, P* - Three datasets based on remote-sensing observations have been selected. All are products calibrated using gauges measurements: the Global Precipitation Climatology Project (GPCP-V2, Adler et al., 2003); the Tropical Rainfall Measuring Mission Multi-satellite Precipitation Analysis (TMPA,3B42-V7, Huffman et al., 2007); and the Multi-Source Weighted-Ensemble Precipitation (MSEWP) dataset Beck et al. (2017).

*Evapotranspiration, E* - Three satellite-based products were chosen to describe evapotranspiration over land. All these
datasets are assumed to be satellite-based products even if their retrieval algorithms can use auxiliary information and a model. the Global Land Evaporation Amsterdam Model (GLEAM-V3B, Martens et al., 2016; Miralles et al., 2011), the global observation-driven Penman-Monteith-Leuning (PML, Zhang Yongqiang et al., 2016) evapotranspiration introduced by the Commonwealth Scientific and Industrial Research Organisation (CSIRO); and the MODIS Global Evapotranspiration Project (MOD16, Mu et al., 2011) will be used due to their different equations of parametrization for the evapotranspiration.





*Total Water Storage Change (TWSC), $\Delta S$* - The TWSC estimates are all based on the GRACE satellite measurements (Tapley et al., 2004). These estimates include the surface (wetland, floodplains, lakes, rivers and man-made reservoirs), soil moisture, snow pack, glaciers and groundwater waters. Satellite datasets are based either on the spherical decomposition of GRACE(for instance (JPL, Watkins and Yuan, 2014) or on the "MASCON" solution: the Jet Propulsion Laboratory (JPL, Watkins et al., 2015a, MSC) product that also includes a scaling factor for hydrological coherency. Based on preliminary tests, it was observed

that the MASCON solution for TWSC ($\Delta S$) was in better agreement with the three other water component estimate, and in particular over the Irrawaddy basin, compared to the spherical solutions. This could be due to the local inversion in MASCON solution that prevent from the de-striping processing usually done in the spherical decomposition of GRACE. It as been shown that de-striping could limit the capability of spherical solution over particularly South/North oriented basin (Wahr et al., 2006; Rateb et al., 2017). In the following, the MASCON solution from JPL is used. Fig. 2 represents the GRACE TWSC and TWSC

anomaly (with respect to averaged season), over the five basins of the previous section. The annual cycle is well pronounced in each basin, showing the strong seasonality of the WC in these regions. The anomalies have strong inter annual variations showing the evolution of the WC along the years.

    *Discharge, $D$* - The Global Runoff Data Centre (GRDC) gathers discharge measurements at the global scale. However, for large tropical rivers, and more particularly over South Asia, only few stations are available and they are not all updated to

recent periods. In particular, among the five considered rivers of this study (Fig. 1), four of them are not available at GRDC and we obtained them instead from personal communication and sharing of local colleagues (Table 1).

    In the following, an *a priori* specification of the uncertainties for each one of these satellite sources are required. For the three precipitation datasets, we specify a 14 mm/month STD (STandard Deviation) error, this corresponds to a 8 mm/month STD for their merging (Aires, 2014). Similarly, the for three evapotranspiration datasets, we specify a 7 mm/month STD,

which corresponds to 4 mm/month for their merging. River discharge is an *in situ* measurement so a 3 mm/month STD is used. Since the objective is the reconstruction of the GRACE observations over long time series, we specify a small uncertainty (1 mm/month) to avoid changing these values during the integration.

### 2.2.2   Datasets used in the evaluation

• *ISBA hydrological model* - To evaluate our reconstruction of the long-term evolution of TWSC over large Himalayan basins,

we also use the ISBA-CTRIP numerical land surface system. ISBA-CTRIP is a "state of the art" hydrological system that simulates TWSC at the global scale with an good accuracy as shown in Decharme et al. (2019). The ISBA-CTRIP TWSC all water mass variation (river water mass and floodplains, snowpack, canopy water, total soil moisture and groundwater storage). The ISBA land surface model explicitly solve the energy and water budgets at the land surface at any time step. The CTRIP river routing model simulates river discharges up to the ocean from the total runoff computed by ISBA. A two-way coupling

between ISBA and CTRIP allows to account for, (1) a dynamic river flooding scheme with explicit interactions between the floodplains, the soil and the atmosphere (through free-water evaporation, precipitation interception and infiltration) and (2) a two-dimensional diffusive groundwater scheme to represent unconfined aquifers and upward capillarity fluxes into the super-ficial soil. More details can be found in Decharme et al. (2019). In this study, we use a product derived from a global offline





simulation at 0.5° resolution done with this ISBA-CTRIP configuration and driven at a 3-hourly time scale by the ERA-Interim

reanalysis over the 1979-2015 period. To ensure that realistic precipitation are fed to the ISBA-CTRIP system (Szczypta et al.,
2014), the ERA-Interim precipitation is, here, hybridized to match the monthly values from the gauge-based Global Precipitation Climatology Center (GPCC) Full Data Product V6 (Schneider et al., 2011, 2014). At each time step, ISBA-CTRIP gives the variation of the total mass of water. The TWSC estimate from ISBA-CTRIP is then the monthly average of this field, which is slightly different than the reconstruction via Eq. (1) (see Appendix A). Since GRACE data are anomalies relative

to a reference geoid, the TWSC estimate from ISBA-CTRIP is also calculated in terms of anomaly over the analysed period. To be consistent with the GRACE data, the simulated TWS were smoothed using a 200 km-width Gaussian filter which is quasi-similar to the filter used for the GRACE products (Watkins et al., 2015a). In the following, ISBA-CTRIP is shortened in ISBA.

• *GLDAS hydrological model* - For comparison purpose, we also use the Noah 2.7.1 land hydrology model of the Global Land Data Assimilation System (GLDAS). Its purpose is to ingest satellite- and ground-based observations using advanced land surface modeling and data assimilation techniques, in order to generate optimal fields of land surface states and fluxes. GLDAS is an uncoupled land surface modeling system that drives multiple models runs globally at the resolution of 0.25°,

and produces results in near-real time. The GLDAS system is described in (Rodell et al., 2004). GLDAS is a platform of assimilation and differs from hydrological models. In particular, they do not model reservoirs. For our comparison, we use the Land Water Content output of GLDAS.

### 2.3   Methodology

The notations are presented in this section but more methodological details can be found in Aires (2014),. The last version of

the integration methodology is well described in Pellet et al. (2018b).

#### 2.3.1   Water cycle budget closure at basin scale

The first step of the integration consists in merging the various datasets presented in Section 2.2.1. The three precipitation datasets are merged based on their respective distance to the mean. A similar approach is used for the three estimates of the evapotranspiration. Only one discharge dataset is available and only the MASCON-JPL is used for TWSC. We denote

$X_{SW} = (P_{SW}, E_{SW}, D_{SW}, \Delta S_{SW})$ where $SW$ stands for "Simple weighting" the results of the merging.

Following Aires (2014), it is then possible to write the conservation of water mass at the basin scale as a constraint applied on the state vector $X = (P, E, D, \Delta S)$. The WC budget constraint is expressed in Eq. (2). A relaxed constraint is considered (Pellet et al., 2018b): the WC budget is closed within an error $r$ that follows a normal distribution with specified uncertainty





(Yilmaz et al., 2011). The problem can be written in the following way:

$$X^t = (P, \ E, \ D, \ \Delta S)$$

$$G = [1, \ -1, \ -1, \ -1] \tag{2}$$

$$X^t \cdot G^t = r \text{ with } r \sim \mathcal{N}(0, \Sigma),$$

where $^t$ is the transpose sign. $G$ the closure operator and $\Sigma$ the variance of the relaxation $r$. The optimised solution of this problem can be expressed as:

$$X_{PF} = (I - K_{PF} \cdot G\Sigma^{-1}G^t) \cdot X_{SW}, \tag{3}$$

where $K_{PF} = (B^{-1} + G\Sigma^{-1}G^t)^{-1}$, $PF$ stands for the "Post-Filtering" of the previous solution $X_{SW}$, and $B$ is the error covariance matrix of $X_{SW}$ that is specified here as (Section 2.2.1):

$$B = \begin{pmatrix} 8 & 0 & 0 & 0 \\ 0 & 4 & 0 & 0 \\ 0 & 0 & 3 & 0 \\ 0 & 0 & 0 & 1 \end{pmatrix}. \tag{4}$$

This methodology allows obtaining a solution $X_{PF}$ that closes the WC budget (within the relaxation $r$).

### 2.3.2 The calibration for the temporal extension of the closure constraint

An important limitation of the closure integration is the need for a common coverage period for all sources of information used in the integration. The optimised dataset cannot be provided for time steps with a missing component. To overcome this issue, a calibration to correct independently each water component towards the closure solution has been introduced (Munier et al., 2014). This calibration is based on a statistical regression between the merged observations $X_{SW}$ and the optimised estimates $X_{PF}$, assuming that this optimised dataset represents the reference. In Pellet et al. (2018b), the calibration is not strictly linear in order to avoid correcting null water fluxes (Munier and Aires, 2017). The following regression is used for $P$, $E$ and $D$:

$$X_{CAL} = a \cdot X_{SW} + b \cdot (1 - e^{\frac{-X_{SW}}{c}}) \tag{5}$$

so that $X_{CAL}$ becomes closer to $X_{PF}$. This calibration is close to a linear calibration, but zero-values are kept unchanged. $a$, $b$ and $c$ are the calibration parameters. The calibration is performed not only during the GRACE period (2002-2015) but over the complete record of each satellite dataset. It was shown in previous studies that the calibration does not allow for a perfect balance of WC, but it greatly reduces the WC budget residuals compared to the original estimates.

The calibration of Eq. (5) is applied independently on each dataset of Section 2.2.1. Table 2 shows the original ($X_{SW}$) versus the calibrated ($X_{CAL}$) Root Mean Squares of the WC budget residuals. It can be seen that the calibration is a significant improvement in each basin, with a decrease of the error from 25 to 54%.

Fig. 3 compares in row the original ($X_{SW}$) and calibrated ($X_{CAL}$) estimates of the four water components with the WC budget residuals, for the Ganges and the Brahmaputra basins, over the GRACE period. It can be seen that for the Ganges,





the water components are not particularly impacted by the calibration. This is due to the overall coherency of the various water components estimates and the relatively low WC budget residual for this particular basin. Only a small improvement can be noticed in the WC budget residuals. For the Brahmaputra basin, precipitation and evapotranspiration are much more impacted by the calibration. The discharge is slightly changed because we specified a low uncertainty on this *in situ* variable (i.e. STD=$1mm/month$). The resulting WC budget residuals are much smaller for the calibrated solution meaning that this solution is more coherent hydrologically.

### 2.3.3 SAtellite Water Cycle (SAWC) reconstruction

Similarly to Munier et al. (2014), the three available water components ($P$, $E$, $D$) are used to estimate the fourth one, the TWSC $\Delta S$. This allows extending temporally the monitoring of the TWSC before and after the GRACE period (2002-2015). In addition, GRACE satellite is down for maintenance every six month since 2011. The calibration approach allows filling gaps measurement in GRACE observation with high accurancy (Munier et al., 2014). Following Landerer et al. (2010) and to avoid temporal mismatching between GRACE-derived TWSC and the monthly estimate of the other water components, we use a centred differences of the mean TWS anomalies to compute the TWSC $\Delta S(t)$. The right-hand side of Eq. (1) is therefore computed for each month $t$ as:

$$\Delta S(t) \quad = \tfrac{1}{4} X'_{CAL}(t-1) + \tfrac{1}{2} X'_{CAL}(t) + \tfrac{1}{4} X'_{CAL}(t+1) \tag{6}$$

where $X'_{CAL} = (P_{CAL} - E_{CAL} - D_{CAL})$, and $\Delta S$ is the centered mass rates:

$$\Delta S_t \quad = \tfrac{S(t+1)-S(t-1)}{2}. \tag{7}$$

### 2.3.4 Uncertainty characterisation

Estimating product uncertainty is a valuable information for instance in an assimilation framework. Most uncertainty estimation approaches require defining first a reference: *in situ*, reanalysis, or a consensus of all available data. In this work, the chosen reference is the optimized product ($X_{PF}$), this is a solution that is hydrologically more coherent and reliable than the original datasets. All satellite datasets are compared to this reference to compute bias (not shown) and uncertainty (standard deviation) errors. For instance, for precipitation: $\sigma_P^2 = E[(P - P_{PF})^2]$. Such an approach was used in Munier and Aires (2017); Pellet et al. (2018b).

Table 3 gathers the uncertainty estimates for all the original satellite datasets, for $P$, $E$ and $D$, over the five considered basins. These *a posteriori* uncertainty estimates are in line with the specifications that were taken *a priori* for each of the datasets (Section 2.2.1). It can be seen that the Brahmaputra has higher uncertainties, especially for precipitation. MSEWP appears less reliable than GPCP or TMPA for precipitation; and GLEAM seems more reliable than MOD16 or CSIRO over these five basins.

It is possible to compute the SAWC reconstruction of TWSC based on Eq. (8). Once calibrated, $P$, $E$ and $D$ estimates are available over a long time period. They can be used to infer $\Delta S$ using the WC budget equation (Eq. (1)), before and after the





GRACE period:

$$\Delta S_{SAWC} \quad = \quad P_{CAL} - E_{CAL} - D_{CAL}. \tag{8}$$

The reconstruction of TWSC has a different temporal coverage for the five basins because $D$ availability varies (see Table 1). The measurement errors of $P$, $E$, and $D$ of Table 3 are assumed to be independent and normally distributed. In this case, the error in the SAWC reconstruction of $\Delta S$ is given by:

$$\sigma_{\Delta S}^2 \quad = \quad \sigma_P^2 + \sigma_E^2 + \sigma_D^2, \tag{9}$$

where $\sigma_P$, $\sigma_E$, $\sigma_D$ are the uncertainties merge estimate from the values of Table 3. For instance:

$$\sigma_P = \frac{1}{4} \sum_{i=1}^{3} \left( \frac{\sum_{k \neq i}(\sigma_k)^2}{\sum_k (\sigma_k)^2} \right)^2 \sigma_i^2. \tag{10}$$

where $\sigma_k$ is the uncertainty estimate of $k^{th}$ precipitation product.

## 3   SAWC reconstruction of TWSC and evaluation

### 3.1   Evaluation over the GRACE period (2002-2015)

The resulting SAWC time series can be observed (red) and compared to GRACE measurements (blue) over the 2002-2015
period in Fig. 2, over the five basins, for the raw and the anomalies. The ISBA simulation is represented too (green). The seasonality is well represented in every estimate. The specific seasonality of each basin is well characterised by the SAWC reconstruction, see for instance the difference between the Mekong and the Brahmaputra seasons. The SAWC reconstruction uses the GRACE data to calibrate the other datasets, but once the $P$, $E$ and $D$ calibrations are done, the SAWC data in Eq. (8) does not use the GRACE data anymore. This is a good demonstration that GRACE-compatible TWSC estimates can
be obtained from the other water components. The rich inter-annual signal in the anomalies is well captured too by SAWC times series. Some extreme years are well captured: e.g. the high extremes of the 2008 year over the Ganges basin, which are depicted also by the ISBA model.

   In order to quantify the agreement of these times series, Fig. 4 represents the correlation, the correlation of the anomalie, and the Root Mean Square of the Difference (RMSD) between the four estimate (GRACE, SAWC, ISBA and GLDAS), over
the five Himalayan basins, for the GRACE period (2002-2015). It can be see that the SAWC times series is highly correlated (0.956 on average) to the GRACE data, better than the ISBA (0.938); GLDAS has much lower agreement with GRACE (0.91) because it misses completely the season in the Irrawaddy basin for some years (not shown for clarity in Fig. 2). Again, it is not surprising that SAWC is close to GRACE because it has been designed to do that. In terms of anomalies, SAWC results are very close to ISBA, except for the Brahmaputra. This will be analysed below. Again, GLDAS seems to have more difficulties over the Irrawaddy basin. The RMSD statistics are better for the SAWC (19 mm/month error) than for ISBA (26 mm/month error), but this is no surprise because the season is a large part of these discrepancies.





Overall, it can be concluded that SAWC seems closer to GRACE than ISBA, for some events, as seen in the anomalies, occurring in the Brahmaputra basin. GLDAS has a lower agreement than ISBA, in particular over the Irrawaddy basin.

### 3.2    Comparison with ISBA TWSC

In Fig. 5, the SAWC reconstruction is compared to the ISBA simulation over a long time record. ISBA is available from 1980 to 2015, SAWC is available based on the river discharge *in situ* measurements coverage (see Table 1). The first important

remark to be done is the very good seasonal cycle agreement between SAWC and ISBA: correlations are larger than 0.93, except for the Brahmaputra with a correlation of 0.88. This basin presents a particular water cycle in 2007 that is analyzed in Appendix (Fig. A3). In the following, the year 2007 has been removed from the comparison analysis. For the Ganges basin, the main difference between the two estimates is that ISBA represents larger negative seasonal peaks and a slight phase in the seasons over the Mekong. In terms of seasonal anomalies, the agreement is also satisfactory with correlations between 0.61

(Godavari and Irrawaddy) to 0.78 (Mekong that is always well represented in our analysis), except again for the Brahmaputra (0.22 correlation). However, based on the 2007 analysis over the Brahmaputra (Fig. A3), we believe that the SAWC anomalies might be more reliable because they use measured *in situ* $D$ compared to the models.

In Fig. 6, we analyse the long-term TWSC time series in terms of anomalies with respect to the climatological season. Furthermore, the times series of these anomalies have been smoothed using a three-year moving window. For instance, a

peak value of 10 mm/month means that the time series was on average 10 mm higher than the climatological season, for 3 consecutive years (i.e. 360 accumulated mm in three years).

In general, SAWC reproduces well the long-term anomalies of the MSEWP precipitation dataset. This satellite dataset was used as input with two close other products (TMPA and GPCP calibrated using the same precipitation gauges) for the SAWC reconstruction (while ISBA uses a mix between GPCC and ECMWF reanalysis, see section 2.b.2). In general, ISBA

precipitation inputs has some differences with MSEWP during the 80's and in 2010-2015, this requires further investigations beyond the scope of this study. The evapotranspiration anomalies are relatively flat for all basins, except for the Godavari where both SAWC and ISBA are in good agreement. By construction, the discharge $D$ measurements are well reproduced by SAWC, but some significant differences can be observed for the ISBA model. These important temporal variations of the $D$ anomalies have an important impact on the other WC components of the SAWC reconstruction. TWSC anomalies $\Delta S$ have a

rather constant behaviour in the ISBA analysis, but large variations are present in the SAWC reconstruction. For instance, there is a large water deficit in the 1990-1991 over the Brahmaputra, or over the Ganges in 1985-1887.

From this comparison, the following conclusions can be drawn. When precipitation from SAWC matches well precipitation used to force ISBA, discharge simulated by ISBA is quite close to *in situ* measurements (discharge from SAWC), as for the Mekong and the Godavari basins, which could be seen as an indicator of the good quality of $P_{SAWC}$. On the contrary, main

differences between $D_{ISBA}$ and $D_{SAWC}$ are either due to large differences between precipitations or to the TWSC dynamics. In ISBA, the groundwater storage is a simple delayed reservoir (with a constant delay parameter) which tends to attenuate the river dynamics. It is then not able to simulate long term groundwater dynamics (Pedinotti et al., 2012). Moreover, the ISBA model does not represent anthropogenic factors such as groundwater extraction, river regulation or irrigation, which may





significantly impact river discharges. Integration of such processes are currently under development, as well as alternatives like
data assimilation (Emery et al., 2018; Albergel et al., 2017).

### 3.3 Indirect evaluation using GIEMS inundation area

An important component of the TWS corresponds to the surface waters. The GIEMS (Global Inundation Extent from Multi-Satellite) database provides an estimation of the inundation extent from 1993 to 2007, at the global scale, on a 0.25° resolution equal-area grid (Prigent et al., 2007). GIEMS was fully assessed over Asian basins, especially using GRACE data (Papa et al.,
2008). The SAWC reconstruction of TWSC and the inundation area time series are represented jointly in Fig. 7 to measure their coherency. Since surface waters are only one part of the TWS, we do not expect a perfect match between the two times series. However, the coherency between them is noticeable, correlation ranges from 0.76 (Godavari) to 0.85 (Ganges). Furthermore, the inter-annual variability of both times series can be measured by the seasonal anomalies, their correlations are significative; they oscillate from 0.24 (Irrawaddy) to 0.42 (Ganges) except for the Brahmaputra where problems were already noticed (see
Fig. 6). This comparison is not a direct evaluation of the TWSC, but the fact that coherency can be found between two completely independent measurements on the water cycle is a positive point for the evaluation of the SAWC reconstruction.

### 4  Conclusion and perspectives

The Total Water Storage (TWS) and its Changes (TWSC) is a crucial element of the water cycle because of its impact on water management, and its role of tracer of human activity. The first measurement available to monitor it came from the GRACE
instrument in 2002. Longer time records being necessary for climate studies, we proposed here to use satellite observations for precipitation and evapotranspiration with *in situ* river discharge measurements to estimate the TWSC over the period 1980-2015. Our approach is based on the water conservation rule over each basin. We performed this task over five major Asian basins because their evolution is related to important questions about water management, climate change, and land-use changes. Our SAWC reconstruction of TWSC has been evaluated using (1) GRACE observations (over the 2002-2007 period), (2) ISBA
model simulations (1980-2015), and (3) surface inundation area (1993-2007). The seasonality and inter-annual variability of SAWC's TWS appear coherent with these independent sources of information.

    The advantages of the proposed methodology are numerous. It is an integration method that gathers all the observations available, satellite and *in situ* measurements. Contrarily to traditional assimilation, this methodology does not use any land or hydrological model, except the water conservation law. It uses the multiplicity of observations to reduce uncertainties on
each one of the water cycle components, and introduces more hydrological coherency among them. If the river discharge measurement is available, it allows to handle a true anthropized discharge (not an idealized natural one, as provided by models in most cases). However, if this discharge is not available, the methodology cannot be applied; and if important uncertainties on discharge measurements are present, these errors will be propagated to all the other water components.

    We foresee many perspectives for this work. First, we would like to extend the TWSC estimation to other basins. This
work can be done over large basins (compatible with the GRACE spatial resolution) and where the *in situ* river discharge



measurements are available. Once this is done over a sufficient number of large basins, the optimized databases can be used as a reference to calibrate the satellite estimates at the global scale. This allows for the use of the satellite observations at the global scale and not only over the basins where the integration was performed (Munier and Aires, 2017). River discharges could potentially be estimated over un-monitored basins, or over longer time series than the monitoring. Total water storage

could also be estimated over monitored basins, over longer times series than the GRACE record (as it was done here). When discharge is not monitored, the use of modeled river discharges could be attempted.

Our methodology can also be used to detect incoherencies in our estimations of the water cycle components. For instance, large budget errors could indicate regions where the evapotranspiration is biased (e.g. due to an under- or over-estimation of the irrigation as in the Nile basin). Our approach could detect such problems and propose a bias-correction of the incriminated

water component.

Finally, we expect to use a similar methodology over connected sub-basins. It is possible to estimate the surface water storage using water extent and topography (Papa et al., 2015), but the horizontal underground transport of water cannot be measured so far. The difference of total water storage and surface water storage should help us estimate the ground water storage and characterise its horizontal transport. This would be a major achievement as this important process of the global water cycle is

largely unknown so far.

## Appendix A: Computation of Total Water Storage Change (TWSC)

For a given month, the TWSC corresponds to the variation of the Total Water Storage (TWS) between the first day of the month and the first day of the following month. As show in Eq. (1), TWSC equals the sum of inflows into the domain ($P$) minus outflows out from the domain ($E + Q$) during the whole month ($P$, $E$ and $Q$ represent monthly averages). The ISBA

land surface model may provide all variables, including TWS, at a daily time step, so that it is possible to compute TWSC as the above-mentioned difference. By construction, the water balance is closed in the ISBA model, and the absolute difference between TWSC and ($P - E - Q$) does not exceed $10^{-6}$ mm/month (with a RMS of $10^{-9}$ mm/month). On the other hand, it is not possible to compute the exact TWSC with GRACE data since TWS at the beginning of each month is not available. Instead, GRACE data correspond to monthly averages of TWS anomalies (temporal mean removed). To approximate TWSC,

we used the centred difference from Eq. (7). Yet, this approximation introduces important errors compared to the true TWSC. Fig. A1 (left) shows the impact of this approximation with ISBA outputs (where the true TWSC is represented by $P - E - Q$). To reduce this error, we followed Landerer et al. (2010) by computing $P - E - Q$ using Eq. (6). Fig. A1 (right) shows a better match between the approximated $P - E - Q$ and the centred TWSC. Nevertheless, the reader should keep in mind that both approximations increase final uncertainties of about 5 to10 mm/month. This error has a quite high frequency and is reduced to

less than 1 mm/month when using a 3-year moving average as in Fig. 7.





## Appendix B: Zoom on 2007 event over Brahmaputra

An extreme inundation has occurred in the Spring of 2007 over the Brahmaputra basin (Gouweleeuw et al., 2018; Islam et al., 2010; Webster et al., 2010). It is interesting here to analyse how this was handled in the SAWC reconstruction and the ISBA model. The Fig. A3 compares them for the four water component estimates. The climatological seasons are also represented (dashed lines) for comparison purpose. Two basins are illustrated: the Ganges and the Brahmaputra.

In this sample, it can be seen that the model follows a classical seasonality for each water component and both basins. In the SAWC reconstruction, the seasonal anomaly is well reproduced for the discharge $D$, which was expected since this observation was used in the SAWC integration process. This translates into a pronounced anomaly in TWSC. This shows the benefits and the risks associated with the SAWC reconstruction: if the *in situ* $D$ is reliable, then SAWC will reproduce it well and the impact on the other components can be important. If $D$ measurements are erroneous, this can introduce considerable noise into the WC analysis.

This relates also to the question of the natural versus real/anthropised discharge $D$. Hydrological model will generally consider natural rivers. It is difficult to obtain all the necessary information to model true discharge (dams management, pumping for irrigation, etc.). An interesting way to constrain models to follow *in situ* measurements of the discharge would be to assimilate these measurements into the model (Wang et al., 2018).

*Acknowledgements.* The authors are grateful to Diego Fernandez for fruitful discussions on the water cycle monitoring during the ESA-WACMOS-MED project.



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



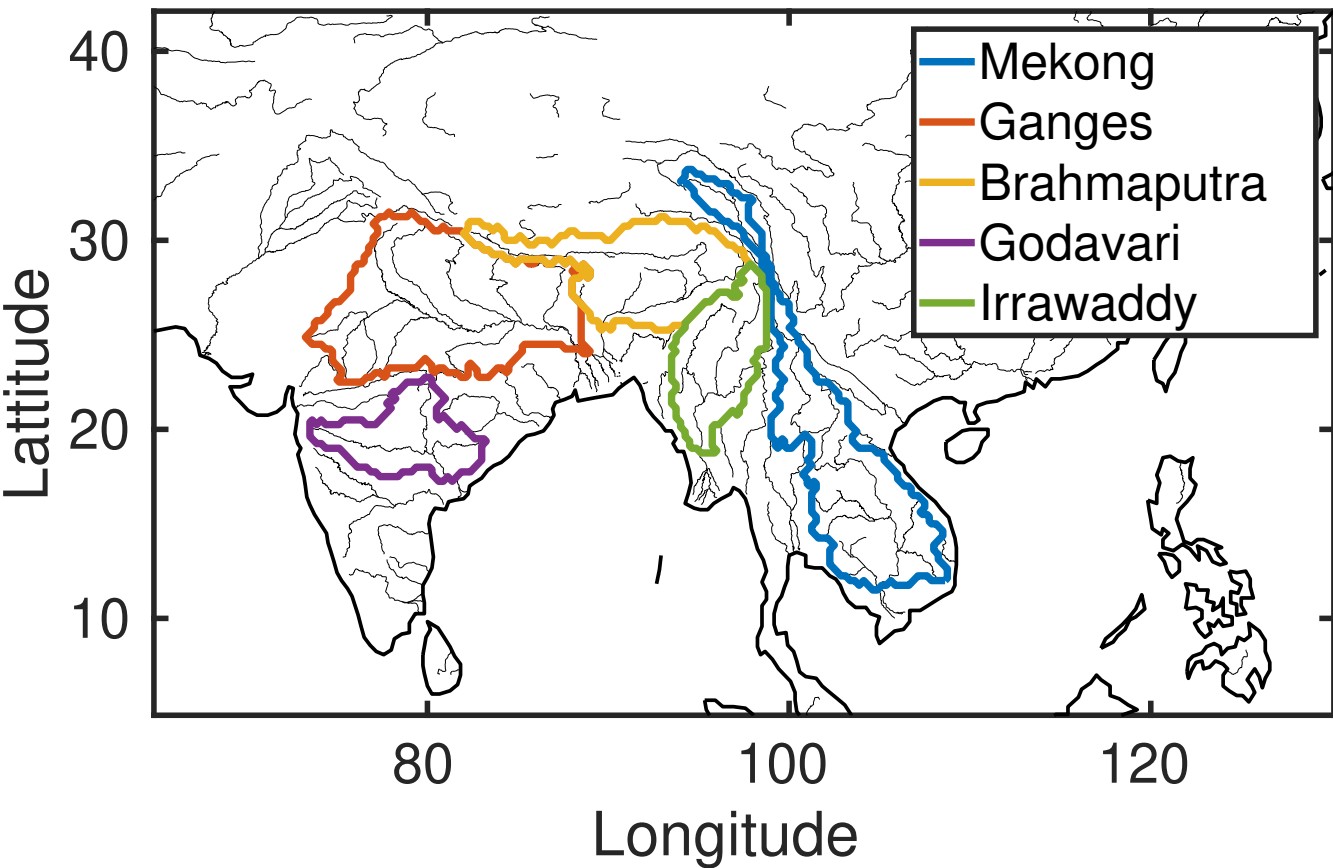

**Figure 1.** Five Himalayan basins considered in this study.





**Figure 2.** TWSC (top) and TWSC seasonal anomaly (bottom), for the three estimates (GRACE, SAWC and ISBA), for the five Himalayan basins, over the GRACE time period.





**Figure 3.** Comparison of the four water components estimates and the WC budget residuals (in row), for the original $X_{SW}$ (blue) and calibrated $X_{CAL}$ (red) estimates. The estimates are for the Ganges (top) and the Brahmaputra (bottom) basins.





**Figure 4.** The correlation (left), the correlation of the anomalies (middle), and the Root Mean Square of the Difference (right) between the four estimates (GRACE, SAWC, ISBA, GLDAS), for the five Himalayan river basins (in row), over the GRACE (2002-2015). Some of the commented statistics are also indicated numerically.

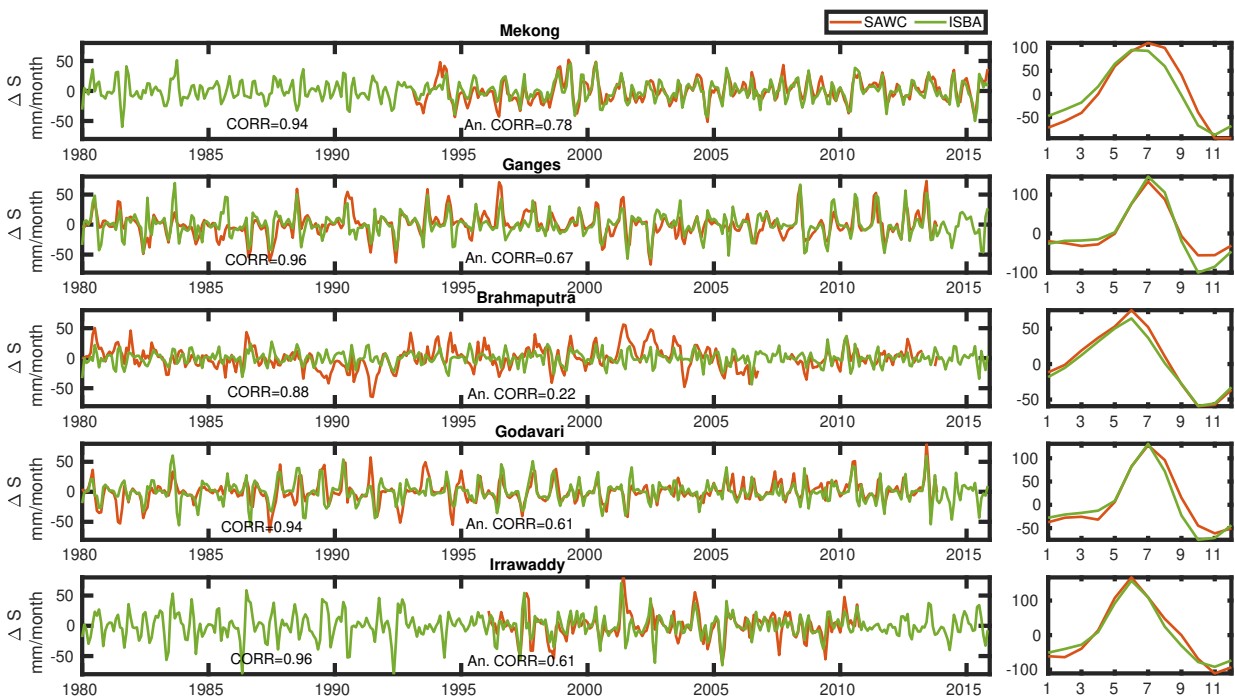

**Figure 5.** Times series of the TWSC (left) and seasonal cycle (right) from 1980 to 2015, for the SAWC and ISBA model estimates, over the five considered basins. Correlations of raw and anomaly values are also provided.





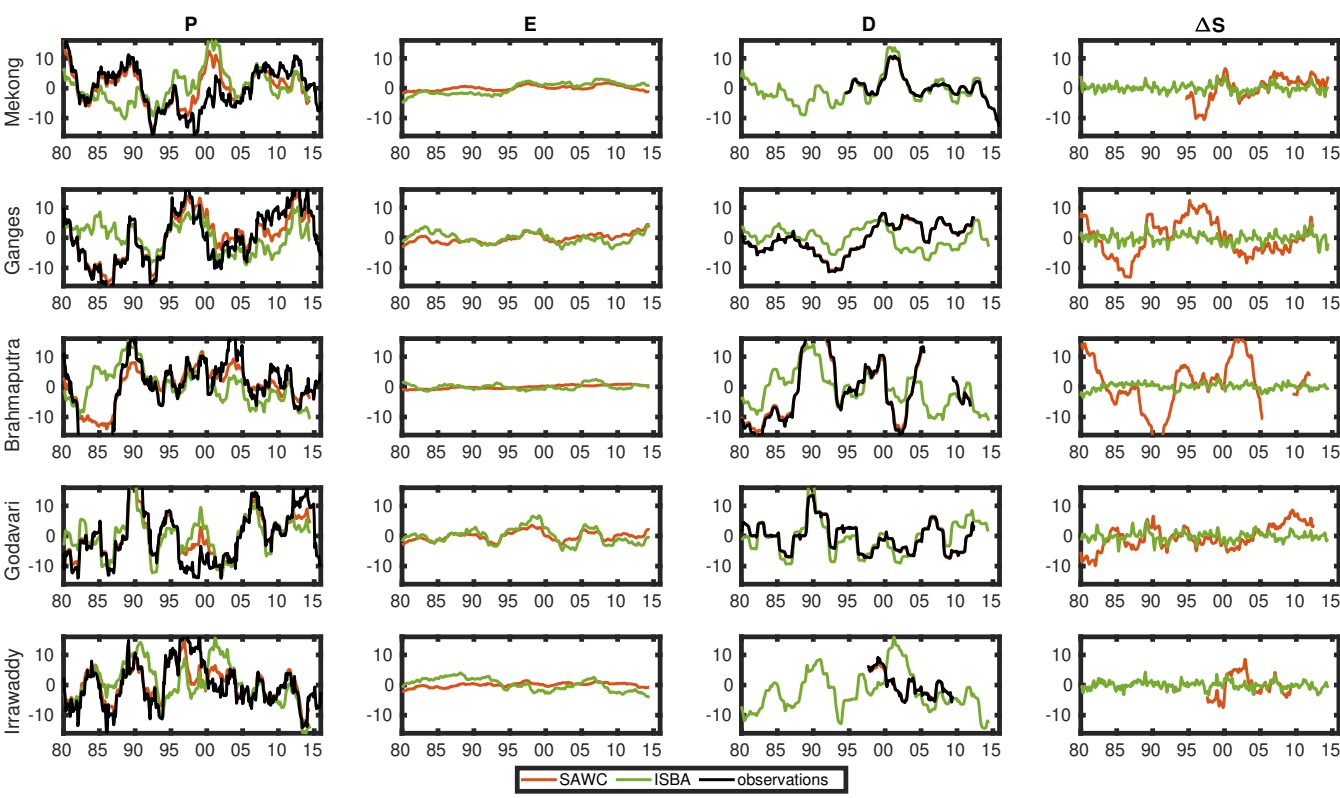

**Figure 6.** Times series of the WC components (mm/month) for 1980-2015, in terms of anomalies (with respect to the climatological season) smoothed using a 3-year moving window; on the five considered basins; for SAWC (red) and ISBA (green) estimates. Observations are also represented in black (MWEWP for $P$ and *in situ* measurements for $D$).





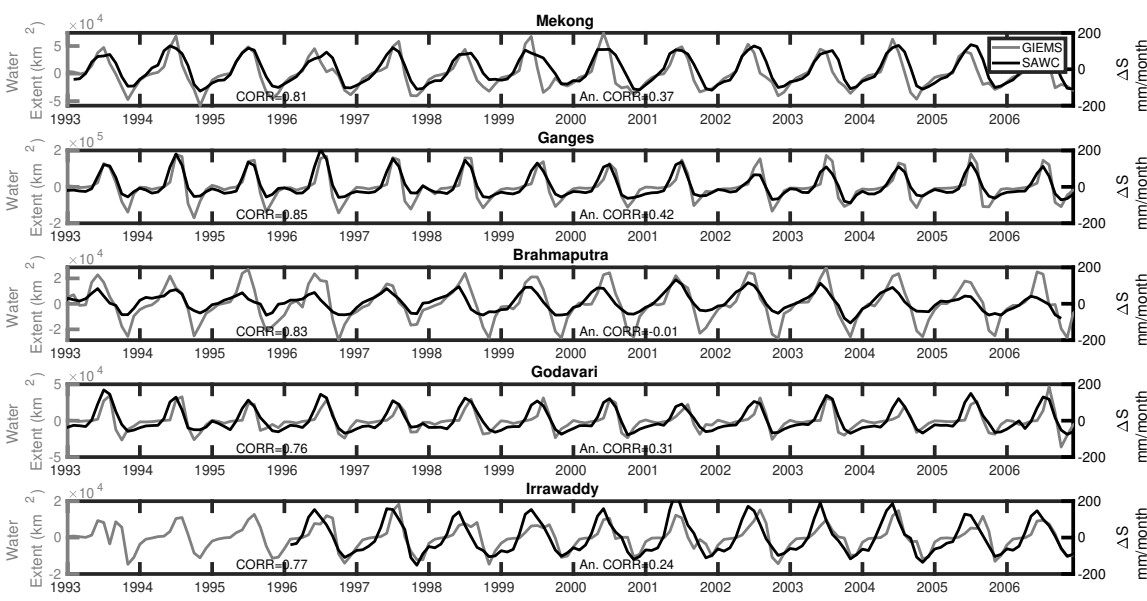

**Figure 7.** Evaluation of TWSC from the SAWC reconstruction using the GIEMS inundated area, from 1993 to 2007, over the figure considered basins. The correlation between them is also provided, together with the correlation of the anomalies.



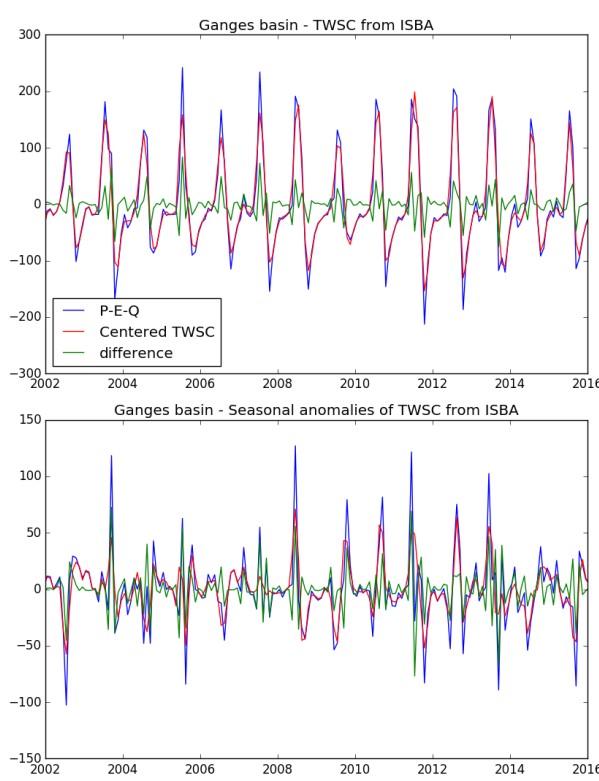

**Figure A1.** Comparison between the two estimates of the TWSC over the Ganges basin: the closure at any time step in the ISBA model ($P$-$E$-$Q$, in blue) and the centered difference of the observed TWS anomalies with GRACE (in red). The difference of the two estimate is also shown (in green). The figure shows the original time series (top) and the seasonal anomalies (bottom).



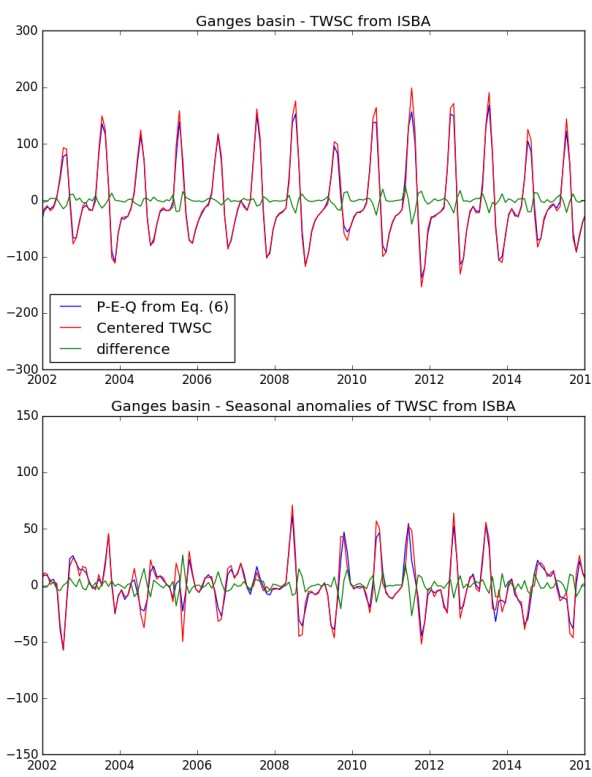

**Figure A2.** Same as Fig.A1 but the closure is now ensured with Eq.(6), following Landerer et al. (2010). This approximation lead to a better match of the two estimates.

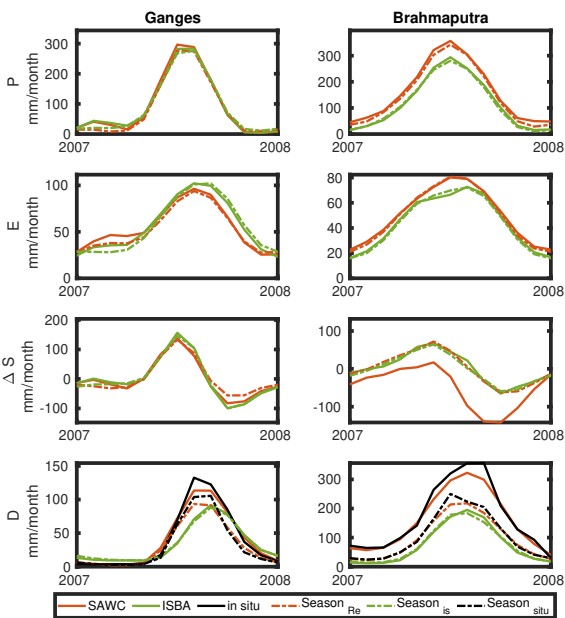

**Figure A3.** Comparison between SAWC reconstruction (red) and the ISBA model output (Green) estimates; for the year 2007 with a large inondation in the Brahmaputra basin; for the four water components estimates (in row); and the Ganges (left) and the Brahmaputra (right) basins. The climatological season are also represented in dashed lines.





| Name | Area (10⁵ km²) | Outlet station | Location | Time record | Source |
|---|---|---|---|---|---|
| Mekong | 7.7 | Phonm Pehn | 11.5°N; 104.9°E | 1993-2016 | pers. comm. (1) |
| Ganges | 9.6 | Hardinge | 24.1°N; 89.0°E | 1980-2013 | pers. comm. (2) |
| Brahmaputra | 5.2 | Bahadurabad | 25.1°N; 89.7°E | 1980-2013 | pers. comm. (2) |
| Godavari | 3.2 | Polavaram | 17.2°N; 81.7°E | 1965-2015 | Water Resources Information System of India |
| Irrawaddy | 3.6 | Pyay | 18.8°N; 95.2°E | 1996-2010 | GRDC |

**Table 1.** Characteristics of the five considered basins and associate *in situ* measurement stations. (1) Personnal communication from Bianca-maria et al., 2017, EGU. Data derived from radar altimetry water level estimations and calibrated against *in situ* measurements following a similar technique as in (Papa et al., 2010) (2) personnal communication and obtained from BDWB (Bangladesh Water Development Board (http://www.bwdb.gov.bd/) as in Papa et al. (2012)

.

| Basin | Original $X_{SW}$ (mm/month) | Calibrated $X_{CAL}$ (mm/month) | Improvement (%) |
|---|---|---|---|
| Mekong | 23 | 16 | 30 |
| Ganges | 20 | 15 | 25 |
| Brahmaputra | 61 | 28 | 54 |
| Godavari | 29 | 20 | 31 |
| Irrawaddy | 44 | 24 | 45 |

**Table 2.** RMS of the WC budget residuals in mm/month for the original ($X_{SW}$) and the calibrated ($X_{CAL}$) estimates, over the five considered basins.

| | Dataset | Basins | | | | |
|---|---|---|---|---|---|---|
| | | Mekong | Gang. | Brahm. | Goda. | Irrawa. |
| $P$ | GPCP | 14 | 15 | 35 | 12 | 10 |
| | TMPA | 14 | 13 | 31 | 13 | 11 |
| | MSEWP | 15 | 17 | 32 | 16 | 19 |
| $E$ | GLEAM | 6 | 5 | 6 | 6 | 6 |
| | MOD16 | 4 | 8 | 6 | 9 | 8 |
| | CSIRO | 8 | 5 | 7 | 5 | 8 |
| $D$ | *In situ* | 3.4 | 2.7 | 4.7 | 2.8 | 7.7 |

**Table 3.** Uncertainty estimates (in mm/month) in terms of STD error compared to $X_{PF}$, for $P$, $E$ and $D$, over the five considered basins, for the original datasets.