# Peer review of "Long-term Total Water Storage Change from a SAtellite Water Cycle (SAWC) reconstruction over large south Asian basins"

_Hydrology and Earth System Sciences, 2019_

## Referee Comment (RC1) · Anonymous Referee #1 · 6 Dec 2019

**General comments:**

In summary, this article tries to reconstruct Total Water Storage Change (TWSC) using satellite-based integrated water cycle components of P and E, and observation-based D in five larger basins in South Asia. Then, TWSC obtained from GRACE, ISBA, and GLDAS were used to evaluate the performance of the reconstructed TWSC here. The topic is interesting, but many attempts have already been made by previous studies (Tang et al., 2017; Humphrey et al., 2017). Major revision is needed before publication, here, a few suggestions that authors should consider while revising are listed below:

First, as shown in Fig. 2 and Fig. 4, SAtellite Water Cycle (SAWC) estimates generally has higher correlation with that from GRACE,ISBA,and GLDAS except for the Irrawaddy Basin. As for the correlation of anomalies, relative higher correlation between SAWC estimates and the other three were found in Mekong and Ganges basins, while much lower correlation was found in the left basins especially in Brahmaputra basin. This highlighted the spatial and temporal variation of the performance of SAWC estimates, therefore, more discussions on such uncertainties are needed.

Second, ISBA was used to evaluate the SAWC estimates due to the long series historical data. However, ISBA model does not represent anthropogenic factors such as groundwater extraction, river regulation or irrigation, which may significantly impact D and TWSC. This is much different from SAWC estimates which might already considered the anthropogenic disturbances. This difference can lead to some big discrepancy as shown in Fig.5 and Fig. 6 for the terms of D and delta S. Therefore, the authors are encouraged to clarify which anthropogenic disturbances have been considered in SAWC estimates and how they affect the discrepancy among different basins in corresponding years. Also, if possible, adding the results from other hydrological models that considers the human activity is highly encouraged.

Third, compared with previous studies of TWSC derive, an integrated utilization of satellite products to retrieve TWSC seems an advantage of this study, but similar idea has been reported in Pan et al. (2012) and Zhang et al. (2016), therefore, clear illustration of the novelty of this study is needed.

Fourth, as shown in B of Eq.4, a priori specification of the uncertainties seems important in obtaining optimized solution through "Post-Filtering", so more explanations of the advanatge for current specification scheme is needed.

**Specific comments:**

Page 5, Line 60-65, this study used one gravity solution based on MASCON-JPL. Other solutions from Center for Space Research (CSR) at the University of Texas at Austin, and GeoForschungsZentrum (GFZ) are available. The comparison of different solutions among different basins are needed to be clarified to support the choice of the solution or using the resembled solution.

Page5, Line 70, "with respect to averaged season", the time period should be specified. Caption of fig. 3, for the original $X_{SW}$ (blue), it should be green as shown in the figure.

Reference:

Humphrey, Vincent, Lukas Gudmundsson, and Sonia I. Seneviratne. "A global reconstruction of climate-driven subdecadal water storage variability." Geophysical Research Letters 44.5 (2017): 2300-2309.

Pan, M., Sahoo, A. K., Troy, T. J., Vinukollu, R. K., Sheffield, J., Wood, and F, E.: Multisource estimation of long-term terrestrial water budget for major global river basins, J. Clim., 25, 3191–3206, https://doi.org/10.1175/JCLI-D-11-00300.1, 2012.

Tang, Yin, et al. "Reconstructing annual groundwater storage changes in a large-scale irrigation region using GRACE data and Budyko model." Journal of hydrology 551 (2017): 397-406.

Zhang, Y., Pan, M., and Wood, E. F.: On Creating Global Gridded Terrestrial Water Budget Estimates from Satellite Remote Sensing, Surv. Geophys., 37, 1–20, https://doi.org/10.1007/s10712-015-9354-y, 2016.

---

## Referee Comment (RC2) · Anonymous Referee #2 · 21 Jan 2020

Review of "Long-term water storage change from a Satellite Water Cycle (SAWC) reconstruction over large south Asian basins" by Pellet et al, submitted to Hydrology and Earth System Sciences.

This paper explains how to estimate the total water storage change of a large basin using GRACE estimates by satellite. The water conservation equation is used to have an independent constraint, and uses satellite estimates of precipitation and evaporation together with a direct measure of river discharge near the mouth of the river.  These complementary measures have to be at the monthly scale, as this is the temporal resolution of the GRACE estimates.

The methodology is applied to four large basins in India and Indochina and the methodology is able to produce estimates that compare well with GRACE observations. I find the paper and the methodology interesting and the results of application, since they allow to monitor the water status of large basins with very little in-situ observations (essentially only a discharge measurement is needed). The paper is clearly written and well organized.

My questions, being a meteorologist, are about the determination of the precipitation and evaporation by satellite. In the integration part, three sources are used for precipitation. More than providing the references, nothing is said about the characteristics of these data sets, how are they produced, what are the differences between them, which is the uncertainty for each of them, and how is the total uncertainty obtained. Similarly, more information about the ET databases should be provided.

I believe that the paper would benefit of related precipitation and evaporation maps and a discussion in depth of the uncertainties of the terms of water closure budget (P, ET, D). The last paragraph of subsection 2.2.1, or Table 3, only give the values imposed for the uncertainties, not how they are obtained. Also subsection 2.3.4 is vague on the subject.

On the other hand, ISBA-CTRIP and GLDAS are used as evaluation tools. Being these utilities models themselves, it is unclear if  the results are good enough for validation in this area of the world. More details should be provided about the quality of these models in this region so that it appears legitimate to use it as a validation tool, discussing at least their uncertainties.

Furthermore, having a better description of the rationale in Section 3, more specifically in subsection 3.2, may be of help for the reader. In subsection 3.1 all the available sources (GRACE, SAWC, ISBA and GLDAS) are compared and it is stated that SAWC fits best with GRACE, admitting that it is by construction. Then, in subsection 3.2, it retains ISBA for the further comparison considering that it performs better than GLDAS. In this part a discussion on the uncertainties of all methods is missing.

For a non-specialist, the paper is interesting and the methodology seems powerful.

---

## Author Comment (AC1) · 17 Feb 2020

**Long-term Total Water Storage Change from a SAtellite Water Cycle (SAWC) reconstruction over large south Asian basins**

**Reviewer 1**

**GENERAL COMMENTS**

• **In summary, this article tries to reconstruct Total Water Storage Change (TWSC) using satellite-based integrated water cycle components of P and E, and observation-based D in five larger basins in South Asia. Then, TWSC obtained from GRACE, ISBA, and GLDAS were used to evaluate the performance of the reconstructed TWSC here. The topic is interesting, but many attempts have already been made by previous studies (Tang et al., 2017; Humphrey et al., 2017). Major revision is needed before publication, here, a few suggestions that authors should consider while revising are listed below:**
**-** Thank you for your comments, we hope that the new version of the manuscript is now in a better shape.

The introduction describes better the state of the art in the reconstruction of TWS anomalies (TWSA) and change (TWSC) in the literature and states the novelty of this article : *If classical approaches to retrieve TWS rely on land surface model (Decharme et al., 2019; Tootchi et al., 2018), studies have recently attempted to using statistical model and various climate drivers. For instance, Humphrey et al. (2017) reconstruct the TWSA using a linear regression from precipitation and temperature, while Chen et al. (2019) use an artificial neural network to reconstruct TWSA based on precipitation, temperature and surface variables (e.g. soil moisture and NDVI). Yang et al. (2018) review and compare several statistical methods (linear, random forest, artificial neural network and support vector machine) to reconstruct TWSA from soil moisture, canopy water and snow water equivalent. These studies focus on the TWSA without monitoring the whole WC.*

*If statistical methods offer the opportunity to estimate TWS anomalies at global scale in a simpler way than the LSM, they do not consider the water balance and the related TWS estimate may not be coherent with the other water components. The water balance at basin scale has been used to estimate TWSC using satellite observations. For instance, Tang et al. (2017) use the Budyko model to estimate annual TWSC based on P and E. A more sophisticated method has been developed where satellite observation were assimilated in the Variable Capicity Model (VIC) at basin scale (Pan et al., 2012) and at the $0.5°$ LSM pixel (Zhang et al., 2017). In order to obtain a TWSC estimate independent of the LSM, another framework has been developed (Aires, 2014; Munier et al., 2014; Pellet et al., 2019). It is based on an integration of satellite observations and in situ river discharge measurement, using the conservation of the water as a constraint, to optimize all sources of information. It has been shown that using the constraint on the observations gives as good results as with the assimilation framework (Munier et al., 2014). We follow here on this framework to reconstruct TWSC with good accuracy.*

● **First, as shown in Fig. 2 and Fig. 4, SAtellite Water Cycle (SAWC) estimates generally has higher correlation with that from GRACE, ISBA and GLDAS except for the Irrawaddy Basin. As for the correlation of anomalies, relative higher correlation between SAWC estimates and the other three were found in Mekong and Ganges basins, while much lower correlation was found in the left basins especially in Brahmaputra basin. This highlighted the spatial and temporal variation of the performance of SAWC estimates, therefore, more discussions on such uncertainties are needed.**

**-** Thank you for this remark. Indeed, the performance of all TWSC estimates varies spatially and numerous factors explain these differences mainly the accuracy of GRACE estimate in terms of seasonal anomalies over the small sub-basins and the large uncertainty of the evapotranspiration estimates used in SAWC. This is now clearer in the text: *SAWC estimate has generally high correlation values with GRACE, ISBA and GLDAS estimates (except for the Irrawaddy Basin). In terms of correlation of anomalies, SAWC estimate is always closer to ISBA than to GRACE even if SAWC has high correlation of anomalies with GRACE (between 0.69 and 0.79) except over the Brahmaputra basin (0.36). Comparatively, GLDAS estimate is less correlated to GRACE over the four basins (except Brahmaputra basin). The RMSD and RMSD of anomalies show similar pattern than the correlation*

*values over all the basins. GRACE presents relatively low spatial resolution (300 $km^2$ at the equator) that can decrease the accuracy of TWSC anomaly estimate for small basins (e.g. Godavari, Irrawaddy). The smaller the basin is, the larger the gap between SAWC-GRACE and SAWC-ISBA correlation becomes. SAWC estimate is based on precipitation and evapotranspiration obtained at finner spatial resolution than GRACE (0.25°). Therefore, SAWC, as ISBA (at the 0.25° spatial resolution) better represents the anomaly over small basins as far as the precipitation and evapotranspiration are accurate. Over the Brahmaputra basin, the large uncertainty of satellite evapotranspiration products over the mountainous area (see the impact of the calibration for the evapotranspiration estimate over this basin in Figure 3) might impact the SAWC TWSC accuracy and explains why GLDAS and ISBA are better over this basin. This assumption is later confirmed in Figure 6 in which precipitation in ISBA and SAWC are close but the anomalies of E differ. Finally, the discrepancy between simulated TWSC from ISBA and GLDAS can be explained by the different representation of aquifers in these two models. While a two-dimensional diffusive groundwater scheme in ISBA represents unconfined aquifer processes (Vergnes and Decharme, 2012; Vergnes et al., 2012), the Noah land model used in the GLDAS simulations did not include surface and groundwater storage. Therefore, the simulated mean seasonal cycle and the inter-annual variability of the TWSC is improved in ISBA (Decharme et al., 2019). On the contrary, deviations from GRACE TWSC can thus be expected with GLDAS (Syed et al., 2008).*

Based on the results presented in Figure 4, we decided to compare our SAWC solution over the long time period only to ISBA. Nevertheless, none of these models included anthropogenic effects and this is now also discussed (see next comment).

**• Second, ISBA was used to evaluate the SAWC estimates due to the long series historical data. However, ISBA model does not represent anthropogenic factors such as groundwater extraction, river regulation or irrigation, which may significantly impact D and TWSC. This is much different from SAWC estimates which might already considered the anthropogenic disturbances. This difference can lead to some big discrepancy as shown in Fig.5 and Fig. 6 for the terms of D and delta S. Therefore, the authors are encouraged to clarify which anthropogenic disturbances have been considered in SAWC estimates and how they affect the discrepancy among different basins in corresponding years.**

**-** Thank you for this remark. With the use of actual river discharge observations, SAWC estimate considers all anthropogenic effects that impact the river along its path (mainly water withdrawal for irrigation and flow regulation by dams). Several points have been added to the comments on the results in order to discuss the impact of dams construction on the Mekong river discharge in the observation and in the model:

- *In Figure 6, Mekong river discharge anomalies show lower min-max range in the observations than in ISBA. Li et al. (2017) highlight the impact of the construction of the Xiaowan and the Nuozhadu dams starting in 1991. The dam reduces the streamflow in particularly wet seasons and increases the streamflow in particularly dry seasons which lowers the anomaly variations.*

- *D is more correlated to precipitation in ISBA (0.94) than in SAWC solutions (0.63). This shows that D in a model is more straightforwardly dependent of the precipitation than in observe state.*

- *On the contrary, TWSC anomaly is less linked to precipitation in the ISBA model than in SAWC solutions where natural recharge is better represent. These difference is also discussed in the Appendix A.*

**• Also, if possible, adding the results from other hydrological models that considers the human activity is highly encouraged.**
**-** Significant efforts that have been made during the last two decades to incorporate anthropogenic impacts in LSM (Hanasaki et al., 2006; Haddeland et al., 2014). These new schemes in LSM are developed offline and mainly at regional scale but significant challenges still remain in their standardization into global LSM as in the availability of the observations (e.g. irrigation, pumping rate) (Pokhrel et al., 2016). Global LSM do not include the global representation of flow regulation and irrigation water needs. Therefore, analyzing the impact of anthropogenic effect into a LSM is beyond the scope of the study. The previous citations and comment have been added to the manuscript in Section 2.2.2.: *"These two global and well known models have been chosen for comparison even if none of them included anthropogenic effects on the river discharge and groundwater storage. Significant efforts have been made during the last two decades to incorporate anthropogenic impacts in LSM (Hanasaki et al., 2006; Haddeland et al., 2014) but crucial challenges still remain. Most of these new schemes in LSMs have been developed and used offline for regional scale studies and without common and standardized framework (Pokhrel et al., 2016; Döll et al., 2016). At global*

*scale, a state of art does not include the global representation of flow regulation and irrigation water needs."*

• **Third, compared with previous studies of TWSC derive, an integrated utilization of satellite products to retrieve TWSC seems an advantage of this study, but similar idea has been reported in (Pan et al., 2012; Zhang et al., 2017). , therefore, clear illustration of the novelty of this study is needed.**
**-** As now stated in the introduction, the Princeton (Pan et al., 2012; Sahoo et al., 2011; Zhang et al., 2017) and the WATCHFULL / WACMOS-MED initiatives (Aires, 2014; Munier et al., 2014; Pellet et al., 2019) are both based on the combination of numerous satellite information and the physical law of water conservation to optimize the latter. However, the first is based on the assimilation of the satellite information into the VIC model while our approach attempted to be as observational as possible. A study has already compared TWSC reconstruction between Princeton and WATCHFULL initiative over the Mississippi (Munier et al., 2014). This is now indicated in the introduction.

• **Fourth, as shown in B of Eq.4, a priori specification of the uncertainties seems important in obtaining optimized solution through Post-Filtering", so more explanations of the advantage for current specification scheme is needed.**
**-** Characterizing the uncertainties of satellite-retrieved products is a difficult task. These specifications are now clearer in the text: *"Such characterizations are generally product and site specific. Some studies (Pan et al., 2012; Sahoo et al., 2011; Zhang et al., 2017) estimate the a priori uncertainty of particular water components based on the spread among the various estimates (taking the spread of estimates as an estimate of the uncertainties can sometimes be dangerous). In our case, this approach would not take into account the fact that the precipitation estimates are not independent. The value used here are derived from (Munier et al., 2014) in which the authors reviewed carefully the literature on this topic. The partitioning of uncertainty between P and E has however been modified to allow larger uncertainty in P since datasets are dependent in our case. As the objective of the current study is to reconstruct GRACE TWSC, the approach assumes lower errors in GRACE estimate that becomes our reference."*

**SPECIFIC COMMENTS**

• **Page 5, Line 60-65, this study used one gravity solution based on MASCON-JPL. Other solutions from Center for Space Research (CSR) at the University of Texas at Austin, and Geo-ForschungsZentrum (GFZ) are available. The comparison of different solutions among different basins are needed to be clarified to support the choice of the solution or using the resembled solution.**

**-** I may misunderstand the comment. To our knowledge, GFZ does not provide a GRACE MASCON solution but only Spherical decomposition one. The MASCON solutions from CSR and JPL differ in their processing and we choose here the JPL solution because it is more independent of the spherical solutions. This information has been added to the manuscript at Section 2.2.1: *"Another MASCON solution exists : the CSR-MASCON solution. The MASCON solutions from CSR and JPL differ in their processing: while JPL solution is based on the explicit estimation of mass anomalies at specific mass concentration block location using the analytical partial derivatives of the inter-satellite range-rate measurements (Watkins et al., 2015), the CSR developed MASCON solution is first based on a Spherical decomposition of the inter-satellite range-rate measurements that is truncated spatially at the location of mass concentration (Save et al., 2016). The two solutions have been compared to the spherical solutions in terms of uncertainty in both min-max range and trend in (Scanlon et al., 2016; Save et al., 2016). We choose here the JPL solution because it is more independent of the spherical solution."*. If it is admitted that for a Spherical solution (JPL,CSR,GFZ), the use of the ensemble mean (simple arithmetic mean of JPL, CSR, GFZ fields) is the most effective in reducing the noise in the gravity field solutions (Sakumura et al., 2014) this might not be the case for MASCON solution. The community uses independently the JPL-MSC or CSR-MSC solutions (Scanlon et al., 2016). In our preliminary test, the JPL-MSC overperforms the spherical solution over particular latitudinal oriented river basin (e.g. Irrawaddy).

• **Page5, Line 70, "with respect to averaged season", the time period should be specified.**

**-** The time period used to computed the average season (2002-2015) is now specified.

• **Caption of fig. 3, for the original XSW (blue), it should be green as shown in the figure.**

**-** Thank you. Caption has been modified.

**References**

[revised manuscript text omitted]

---

## Author Comment (AC2) · 17 Feb 2020

**Long-term Total Water Storage Change from a SAtellite Water Cycle (SAWC) reconstruction over large south Asian basins**

**Reviewer 2**

**COMMENTS**

• This paper explains how to estimate the total water storage change of a large basin using **GRACE** estimates by satellite. The water conservation equation is used to have an independent constraint, and uses satellite estimates of precipitation and evaporation together with a direct measure of river discharge near the mouth of the river. These complementary measures have to be at the monthly scale, as this is the temporal resolution of the **GRACE** estimates. The methodology is applied to four large basins in India and Indochina and the methodology is able to produce estimates that compare well with **GRACE** observations. I find the paper and the methodology interesting and the results of application, since they allow to monitor the water status of large basins with very little in-situ observations (essentially only a discharge measurement is needed). The paper is clearly written and well organized. - Thank you for your appreciation on this work.

• My questions, being a meteorologist, are about the determination of the precipitation and evaporation by satellite. In the integration part, three sources are used for precipitation. More than providing the references, nothing is said about the characteristics of these data sets, how are they produced, what are the differences between them, which is the uncertainty for each of them, and how is the total uncertainty obtained. Similarly, more information about the ET databases should be provided.

- Thank you for this remark. These global precipitation and ET database are widely used in remote sensing community (Rodell et al., 2015; Pan et al., 2012; Sahoo et al., 2011; Zhang et al., 2017; Munier et al., 2014; Pellet et al., 2018, 2019), but more information are needed in this section. The manuscript specifies some characteristics of these datasets for P, E and TWSC, and cite some inter-comparison studies in which uncertainty assessments can be found:

1. *Precipitation, P - All these datasets are global datasets widely used in the community. GPCP and TMPA use the same algorithm Threshold Matched Precipitation Index (TMPI) to estimate instantaneous precipitation from multiple satellites by combining high-quality passive micro-wave observations and infrared data and differ only in the use of gauge analyses (GPCC) to obtain calibrated estimates. While TMPA is based on inverse random-error variance weighting, GPCP assumes that the precipitation distribution estimated from combined satellite estimates is optimal and uses the gauge observations only for debiasing. The MSWEP dataset merges the highest quality precipitation data sources available as a function of timescale and location. It uses a combination of rain gauge measurements, the two previous satellite datasets, and reanalysis. These datasets have been compared in terms of uncertainties and performance in (Sun et al., 2018). It should be noted that these datasets are not independent of each other but represent the best up-to-date precipitation estimates for hydrological studies.*

2. *Evapotranspiration, E - The Global Land Evaporation Amsterdam Model (GLEAM-V3B, Martens et al., 2016; Miralles et al., 2011), uses Priestley and Taylor 1972 (Priestley and Taylor, 1972) empirical energy-based equation to calculate the reference evapotranspiration and separately estimate the different components of land evaporation: transpiration, bare-soil evaporation, interception loss, open-water evaporation and sublimation. GLEAM uses reanalysis (vA) or satellite (vB) precipitation inputs. The global observation-driven Penman-Monteith-Leuning (PML, Zhang Yongqiang et al., 2016) evapotranspiration introduced by the Commonwealth Scientific and Industrial Research Organisation (CSIRO) and the MODIS Global Evapotranspiration Project (MOD16, Mu et al., 2011) are both evapotranspiration estimates based on Penman-Monteith equations (PENMAN, 1948; Monteith, 1965). We choose these three datasets due to their different equations of parametrization for the evapotranspiration. Inter-comparison of global evapotranspiration algorithms and datasets can be found in (Michel et al., 2016).*

3. *Total Water Storage Change (TWSC), $\Delta S$ - Another MASCON so-lution exists : the CSR-MASCON solution. The MASCON solutions from CSR and JPL differ in their processing: while JPL solution is based on the explicit estimation of mass anomalies at specific mass concentration block location using the analytical partial derivatives of the inter-satellite range-rate measurements (Watkins et al., 2015), the CSR developed MASCON solution is first based on a Spherical decomposition of the inter-satellite range-rate measurements that is truncated spatially at the location of mass concentration (Save et al., 2016). The two solutions have been compared to the spherical solutions in terms of uncertainty in both min-max range and trend in (Scanlon et al., 2016; Save et al., 2016). We choose here the JPL solution because it is more independent of the spherical solution.*

*Characterizing the uncertainties of satellite-retrieved products is a diffi-cult task (see the answer to the next comment). In this study, the precip-itation and evapotranspiration uncertainties are derived from the literature (Munier et al., 2014). All datasets describing a water components have the same uncertainty and the resulting uncertainty of ensemble mean is de-rive assuming the independence of the sources. This simplification is usually done: (Pan et al., 2012; Munier et al., 2014; Pellet et al., 2018, 2019). This is now clearer in the manuscript where the Simple Weighting (i.e. arithmetic average) estimate and its uncertainty are introduced in Section 2.2.1.*

$$P_{SW} = \frac{1}{p-1} \sum_{i=1}^{p} \frac{\sum_{k \neq i}(\sigma_k)^2}{\sum_k (\sigma_k)^2} P_i. \tag{1}$$

*This equation is valid when there is no bias error in the $P_i$s (thanks to the preliminary bias correction) and is optimal when the errors $\epsilon_i$ are statistically independent from each other. This expression is valid for the other water components. The variance of the $P_{SW}$ uncertainties is then given by:*

$$\sigma_{P_{SW}} = \frac{1}{(p-1)^2} \sum_{i=1}^{p} \left( \frac{\sum_{k \neq i}(\sigma_k)^2}{\sum_k (\sigma_k)^2} \right)^2 \sigma_i^2. \tag{2}$$

● **I believe that the paper would benefit of related precipitation and evaporation maps and a discussion in depth of the uncertainties of the terms of water closure budget (P, ET, D). The last paragraph of subsection 2.2.1, or Table 3, only give the values imposed for the uncertainties, not how they are obtained. Also subsection 2.3.4 is vague on the subject.**

- Thank you for this comment. Uncertainty analysis at grid scale is beyond the scope of this study which focus on the basin scale, however particular analysis on precipitation (resp. evapotranspiration) uncertainties can be found in (Sun et al., 2018) (resp. (Michel et al., 2016)). These references have been added in section 2.2.1. The following specification are now clearer in the text : *"Characterizing the uncertainties of satellite-retrieved products is a difficult task. Such characterizations are generally product, and site, specific. Some studies (Pan et al., 2012; Sahoo et al., 2011; Zhang et al., 2017) estimate the a priori uncertainty of particular water component based on the spread among the various estimates. In our case, this approach would not take into account the fact that our precipitation estimate are not independent. Finally, the values considered here are derived from (Munier et al., 2014) in which the authors reviewed the literature on this topic. Compared to this study, the partitioning of uncertainty between P and E has been modified to allow larger uncertainty in P since all P datasets are dependent with each others. As the objective of the current study is to reconstruct GRACE TWSC, the approach assumes low error in the GRACE estimate."*

Noted that Table 3 does not give *a priori* uncertainty value but provided uncertainty estimates computed *a posteriori* as the distance between the original datasets and the reference (our new estimate). This is why the various original products have different *a posteriori* uncertainty even if a same *a priori* uncertainty was specified. This is now clearer in the text.

**• On the other hand, ISBA-CTRIP and GLDAS are used as evaluation tools. Being these utilities models themselves, it is unclear if the results are good enough for validation in this area of the world. More details should be provided about the quality of these models in this region so that it appears legitimate to use it as a validation tool, discussing at least their uncertainties.**
- Thank you for this remark. We prefer to state "evaluation" with ISBA instead of "validation" in section 3.2 since validating TWSC over the long time period (1980-2015) is a difficult task. Often, this type of evaluation is performed by comparing to other independent estimates. For instance, Figure 4 shows that ISBA can simulate more accurately the TWSC than GLDAS. The following statement has been aded to the manuscript : *"Finally, the discrepancy between simulated TWSC from ISBA and GLDAS can be explained by the representation of aquifers in these two models. While a two-dimensional diffusive groundwater scheme in ISBA represents unconfined aquifers process (Vergnes and Decharme, 2012; Vergnes et al., 2012),*

*the Noah land model used in the GLDAS simulations did not include surface and groundwater storage. Therefore, the simulated mean seasonal cycle and the inter-annual variability of the TWSC is improved in ISBA (Decharme et al., 2019). On the contrary, deviations from GRACE TWSC can thus be expected with GLDAS simulated TWSC (Syed et al., 2008). Based on the results presented in Figure 4, we decided to compare SAWC estimate only to ISBA over long time period"*. Nevertheless, none of these models included anthropogenic effect and this is also now discussed.

• **Furthermore, having a better description of the rationale in Section 3, more specifically in subsection 3.2, may be of help for the reader. In subsection 3.1 all the available sources (GRACE, SAWC, ISBA and GLDAS) are compared and it is stated that SAWC fits best with GRACE, admitting that it is by construction. Then, in subsection 3.2, it retains ISBA for the further comparison considering that it performs better than GLDAS. In this part a discussion on the uncertainties of all methods is missing.**
**-** See previous answer for the rational in Section 3. The manuscript now justifies the ISBA comparison.

• **For a non-specialist, the paper is interesting and the methodology seems powerful**
**-** Thank you, we hope that the revised manuscript will answer your concerns.

**References**

Decharme, B., Delire, C., Minvielle, M., Colin, J., Vergnes, J., Alias, A., SaintMartin, D., Séférian, R., Sénési, S., and Voldoire, A. (2019). Recent Changes in the ISBACTRIP Land Surface System for Use in the CNRMCM6 Climate Model and in Global OffLine Hydrological Applications. *J. Adv. Model. Earth Syst.*, 11(5):1207–1252.

[revised manuscript text omitted]